# MFCD-Net: Cross Attention Based Multimodal Fusion Network for DPC Imagery Cloud Detection

**Jingjing Zhang** [1,†]**, Kai Ge** [1,†]**, Lina Xun** [1]**, Xiaobing Sun** [2]**, Wei Xiong** [2]**, Mingmin Zou** [3]**, Jinqin Zhong** [4] **and Teng Li** [1,*]

1. Key Laboratory of Intelligent Computing and Signal Processing of Ministry of Education, School of Electrical Engineering and Automation, Anhui University, Hefei 230601, China
2. Key Laboratory of Optical Calibration and Characterization, Chinese Academy of Sciences, Hefei 230601, China
3. Institutes of Physical Science and Information Technology, Anhui University, Hefei 230601, China
4. School of Internet, Anhui University, Hefei 230601, China
* Correspondence: liteng@ahu.edu.cn
† These authors contributed equally to this work.

**Abstract:** As one kind of remote sensing image (RSI), Directional Polarimetric Camera (DPC) data are of great significance in atmospheric radiation transfer and climate feedback. The availability of DPC images is often hindered by clouds, and effective cloud detection is the premise of many applications. Conventional threshold-based cloud detection methods are limited in performance and generalization capability. In this paper, we propose an effective learning-based 3D multimodal fusion cloud detection network (MFCD-Net) model. The network is a three-input stream architecture with a 3D-Unet-like encoder-decoder structure to fuse the multiple modalities of reflectance image, polarization image Q, and polarization image U in DPC imagery, with consideration of the angle and spectral information. Furthermore, cross attention is utilized in fusing the polarization features into the spatial-angle-spectral features in the reflectance image to enhance the expression of the fused features. The dataset used in this paper is obtained from the DPC cloud product and the cloud mask product. The proposed MFCD-Net achieved excellent cloud detection performance, with a recognition accuracy of 95.74%, according to the results of the experiments.

**Keywords:** cloud detection; polarization; multimodal fusion; cross attention

## 1. Introduction

In recent years, owing to the fast development of remote sensing technologies, remote sensing images (RSI) have gained increasing application value in Earth observation [1], resource investigation, environmental monitoring, and protection [2]. Cloud detection is important in RSI processing since most RSI are contaminated by clouds [3–5], which decreases the quality of RSI and influences the subsequent applications [6]. Simultaneously, clouds play a crucial role in Earth's radiation balance, water cycle, and climate change [7]. Rapid and accurate cloud detection can help provide an effective data source for the inversion of cloud and aerosol parameters and the study of sea color characteristics.

Directional Polarimetric Cameras (DPC) have attracted much attention as an emerging Earth observation technology [8]. Compared with traditional optical remote sensing means, clouds and aerosols are more sensitive to polarization information, which makes satellite polarimetric remote sensing more advantageous in atmosphere detection [9]. The main task of DPC is to obtain multi-band and multi-angle polarized radiation and reflection information. They are used in researching the optical and physical properties of atmospheric aerosols, clouds, and marine water observations. They also provide remote sensing data support for global climate change and atmospheric environment monitoring [8,10–12]. The

purpose of this paper is to explore effective cloud detection of DPC imagery, which is the premise of cloud characteristic parameter inversion and ocean water color inversion.

Existing cloud detection methods for polarized RSI, such as DPC imagery, mainly use the difference in reflectance and polarization reflectance between clouds and the ground to set thresholds to detect cloud regions. However, these thresholding algorithms are often limited in performance and generative ability. For example, certain bright surfaces with high reflectance (ice and snow regions, ocean flare regions, etc.) have very small reflectance differences with cloud regions, making threshold setting difficult. Second, different regions often require setting different thresholds. It is, therefore, necessary to investigate more precise and advanced cloud detection algorithms for DPC imagery cloud detection.

Recently, deep neural network-based methods have been used in the field of RSI cloud detection, typically developed based on semantic segmentation networks, such as Fully Convolutional Networks (FCN) [13], U-Net [14,15], and SegNet [16]. However, semantic segmentation, as a pixel-level classification process, generally requires extremely high image resolution. Their applications are generally for high-resolution RSI, whereas the spatial resolution of DPC imagery is extremely low; therefore, these networks are not suitable. This paper aims to explore effective cloud detection in DPC imagery.

DPC imagery differs from other RSI in that it has multi-spectrum, multi-angle, and multi-polarization characteristics. DPC consists of eight bands with nine observations, each from different angles, and three bands are polarized (with additional Stokes vector images Q and U). Hence, DPC can produce 126 observations per pixel, providing rich information for Earth observation [12,17,18]. To jointly utilize the information from multiple angles, we proposed to use 3D convolution [19] to extract and use angle features. Furthermore, the Squeeze-and-Excitation Network (Senet) [20] is applied to automatically extract the essential spectrum information while avoiding losing spectral information during the band selection process. Finally, some previous studies showed that using the polarization information of clouds can detect cloud pixels in certain areas (such as flare areas) [9,12,17,21]. Polarization images, as another modality different from multi-spectral images, can provide additional information for cloud detection. Inspired by that, we propose to fuse the DPC multi-spectral image and polarization image with a multi-stream architecture, where each stream corresponds to a modality.

Consequently, we propose MFCD-Net, a 3D multimodal fusion network based on cross attention, which takes the multi-angle reflectance image, polarization image Q (Stokes vector Q), and polarization image U (Stokes vector U) as inputs for cloud detection. First, we use angle as the third dimension of the 3D convolution to extract the spatial-angle information completely. The use of 3D convolution can greatly improve the ability to represent multi-dimensional data [22,23]. In addition, we consider spectral bands as channel dimensions and assign different weights to each band based on Senet. Furthermore, we use the cross attention [24] fusion technique in the fusion stage, which enhances the expression of the results of fusion between different modalities. Finally, the overall structure of the whole network is similar to 3D-Unet, and the multimodal fusion operation is carried out in four stages in the down-sampling part, which not only improves spatial feature extraction but also strengthens feature fusion representation by repeated fusion.

So far, there is no DPC cloud detection dataset available. To evaluate the proposed method, we use DPC's Level-1 cloud products and corresponding Level-2 cloud mask products as the data and label for our dataset, respectively. The dataset contains 126 observations per pixel of the DPC imagery. The main contributions of this paper can be summarized as follows.

- To the best of our knowledge, ours is the first approach to introduce the notion of multimodal learning for RSI cloud detection.
- The traditional thresholding algorithm has poor performance and limited generalization ability. Furthermore, given the extremely low spatial resolution of DPC images, conventional semantic segmentation-based methods also fail to achieve good performance. To improve the detection performance, this paper proposes a 3D multimodal

fusion network (MFCD-Net). It makes up for the lack of spatial features by extracting and using angle features, spectral features, and polarization features, thus achieving good performance.

- Simple concatenation and summation feature fusion methods can hardly solve the feature fusion problem where there is an imbalance between features in multimodal data and information inequality contained by different features [25]. In order to enhance the feature fusion effect between different modalities, a cross-attention fusion module is designed in this paper. It takes the attention map from the polarization modality to enhance the representation of the reflectance modality.

The remainder of this paper is organized as follows. Section 2 reviews the related work. Section 3 presents the GF-5 DPC and the corresponding cloud detection dataset. Section 4 describes our multimodal network in detail. Section 5 reports the experimental results and discusses the experiment details. Finally, Section 6 offers the conclusion of this research.

## 2. Related Work

Up to the present, many efforts have been made for cloud detection of RSI, and a variety of cloud detection methods have been proposed. These methods mainly rely on spectral information, frequency information, spatial texture, and other information, in combination with thresholding, clustering, support vector machines, neural networks, and other algorithms for detection [26]. Roughly speaking, they can be summarized as rule-based methods and machine learning-based methods.

The earliest and most generally used rule-based cloud detection method is the spectral threshold method [27]. Several spectral threshold algorithms for DPC imagery have been developed. These methods use reflectance or polarization reflectance to define the threshold. For example, JinghanLi et al. [10] proposed a multi-information cooperation (MIC) method. Rather than relying on a single constant threshold, the MIC utilizes dynamic thresholds simulated by multiple atmospheric models, time intervals, and underlying surfaces. Some other rule-based cloud detection methods were developed based on textural features. Gray co-occurrence matrix, fractal dimension, and boundary features are the most widely used among the different texture analysis methods [28] since they are compatible with the texture properties of the cloud. Though straightforward, the rule-based methods have limited generalization capability and are deficient in terms of performance.

Automatically learning from training data, machine learning algorithms such as conventional random forest [29], K-nearest algorithm [28], and support vector machine [30] have been applied to cloud detection algorithms. Owing to their excellent performance in many vision tasks, deep learning algorithms have emerged as the most popular methods for cloud detection. Fengying Xie et al. [31] proposed a cloud detection algorithm that divides an image into super-pixels by improving simple linear clustering (SLIC) and designs a CNN with two branches to extract multi-scale features of each super-pixel to distinguish pixels. More cloud detection algorithms are based on semantic segmentation models. JingyuYang et al. [6] proposed a cloud detection neural network (CDnet) with an encoder–decoder structure, feature pyramid module, and boundary refinement block to detect cloud areas in thumbnails effectively. Zhenfeng Shao et al. [32] superimposed the visible, near-infrared, shortwave, cirrus, and thermal infrared bands of the Lansat8 satellite to obtain complete spectral information and then proposed a convolution neural network based on multi-scale features to identify thick clouds, thin clouds, and non-cloud regions. Weakly supervised cloud detection methods have also been developed. Zou et al. [33] defined cloud detection as a mixed energy separation process of image foreground and background. The generative antagonistic framework is utilized to establish the groundwork for weak supervision of the cloud image by combining the physical principles behind the cloud image. Yansheng Li et al. [34] proposed a weakly supervised deep learning-based cloud detection (WDCD) method that uses block-level labels to reduce the labor required for annotating the pixel-level labels. In the past two years, the research of deep learning in the field of cloud detection has gradually matured. The problems of cloud boundary blurring

and computational complexity have become recent research hotspots. Kai Hu et al. [35] proposed Cloud Detection U-Net (CDUNet), which could refine the division boundary of the cloud layer and capture its spatial position information. To reduce the computational complexity without affecting the accuracy, Chen Luo et al. [36] developed a lightweight autoencoder-based cloud detection network, LWCDNet. Qibin He et al. [37] proposed a lightweight network (DABNet) to achieve high-accuracy detection of complex clouds, with not only a clearer boundary but also a lower false-alarm rate.

Even though the deep learning-based methods discussed above have achieved impressive performance, they are difficult to apply to DPC imagery due to the low spatial resolution of DPC imagery. Therefore, this paper proposes a novel MFCD-Net using 3D convolution, Senet, and cross attention fusion to extract and utilize angle, spectral, and polarization information to compensate for the lack of spatial information and achieve superior performance.

## 3. Dataset

In this section, we detail the description of the dataset. First, we present the data sources and descriptions. In addition, we explain the reasoning behind our data selection. Finally, we discuss how the dataset was processed.

### 3.1. Data Sources and Description

As an important payload of the GF-5 satellite, the Directional Polarimetric Camera (DPC) is the first Chinese multi-angle polarized observation sensor. DPC imagery has the properties of multi-angle, multi-spectrum, and multi-polarization, with a spatial resolution of 3.3 km × 3.3 km and a swath width of 1850 km. DPC has three polarized bands (490, 670, 865 nm) and five unpolarized bands (443, 565, 763, 765, and 910 nm). Table 1 lists the precise details for each band. Based on the satellite platform's ultra-wide (100°) field of view and continuous imaging capability, DPC can view targets from 9 various observation angles, producing an observation vector with at least 126 measurements per pixel (shown in Figure 1).

**Table 1.** The spectral band of DPC.

| Band | Central Wavelength | Band Width | Number of Observations |
|---|---|---|---|
| Band1 | 443 nm | 433~453 nm | 9 |
| Band2 (Polarization) | 490 nm | 480~500 nm | $9 \times 3 = 27$ |
| Band3 | 565 nm | 555~575 nm | 9 |
| Band4 (Polarization) | 670 nm | 660~680 nm | $9 \times 3 = 27$ |
| Bnad5 | 763 nm | 758~768 nm | 9 |
| Band6 | 765 nm | 745~785 nm | 9 |
| Band7 (Polarization) | 865 nm | 845~885 nm | $9 \times 3 = 27$ |
| Band8 | 910 nm | 900~920 nm | 9 |

Our dataset is derived from DPC L1-level cloud products and the corresponding L2-level cloud mask products produced by the Hefei Institute of Material Sciences of the Chinese Academy of Sciences. There are 14 large DPC images and corresponding cloud mask labels with a size of 12,168 × 6084. These images were selected based on cloud coverage and the underlying surface. To enhance the usefulness of the dataset, all images have high cloud coverage, with a small difference in the proportion of the lower underlying surface being ocean and land.

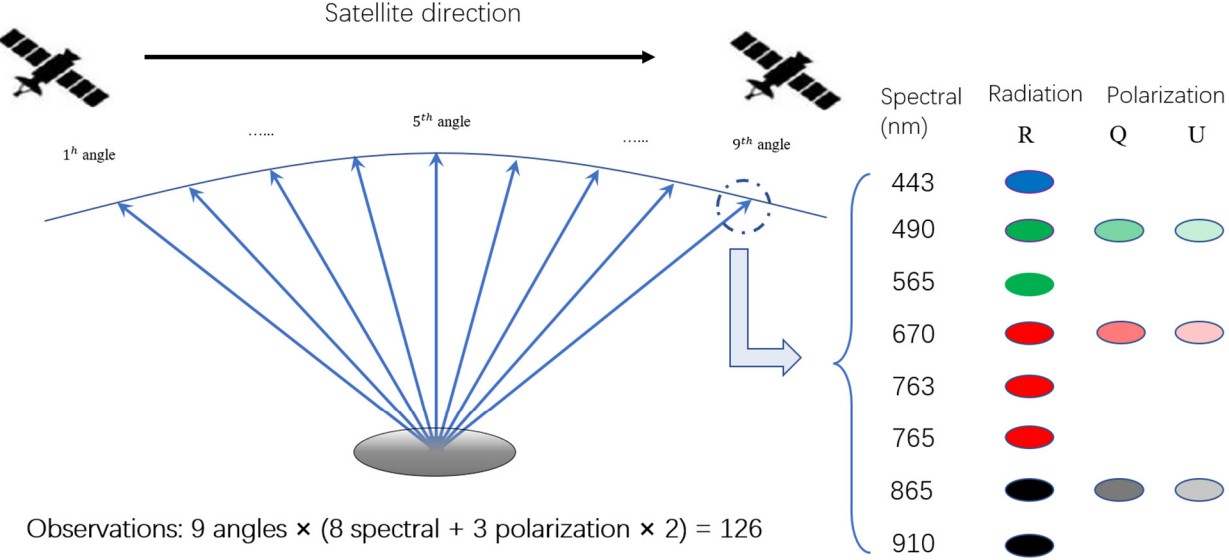

**Figure 1.** The multi-angle observation principle and the description of the measurement dataset of DPC.

*3.2. Data Selection*

Reflectance is the ratio of the reflected energy from the surface of an object to the incident energy reaching the surface of the object. There are many threshold cloud detection algorithms that identify cloud pixels based on reflectance and brightness temperature [38]. This is due to the fact that cloud pixels have higher reflectance than clear sky pixels, and this is also the core detection condition of the traditional threshold method. To deepen the distinction between cloud and non-cloud pixels, we use the DPC reflectance data as part of the dataset. The reflectance data are calculated based on the normalized radiance data and solar zenith angle data in DPC L1-level cloud products by the formula:

$$R = \frac{\pi L}{\mu_s E} \tag{1}$$

where $L$ represents radiation brightness, $\mu_S$ represents the cosine value of the solar zenith angle, and $E$ represents solar irradiance at the top of the atmosphere.

Middle-level clouds, low-level clouds, and thin cloud regions have low reflectance. The difference between their reflectance and surface reflectance is not obvious, and it is difficult to obtain satisfactory cloud detection results only by using reflectance images. There is increasing evidence that polarized radiation information can contribute to improving cloud recognition abilities for these areas [39,40]. Therefore, our dataset also includes polarization radiation data, i.e., Stokes vector images Q and U. Q denotes the intensity of linear polarization parallel or perpendicular to the reference plane, and U denotes the intensity of linear polarization at the angle of 45 degrees to the reference plane.

Considering the high number of spectral bands presented in DPC imagery, the spectral threshold method generally selects some appropriate band's reflectance or polarization reflectance data for threshold detection. The polarization reflectance $R_P$ is generally defined by the following equation:

$$R_P = \frac{\sqrt{Q^2 + U^2}}{\mu_s E} \tag{2}$$

where $Q$ and $U$ represent Stokes vector Q and U, $\mu_S$ represents the cosine value of the solar zenith angle, and $E$ represents solar irradiance at the top of the atmosphere.

In our dataset, however, we have chosen the reflectance image of all eight bands and the polarization image (Stokes vector images Q and U) of all three bands. There are four main reasons:

- There is useful information for cloud identification in the reflectivity and polarization images of each band;
- By using the channel attention mechanism, the important band can be highlighted, the unimportant band can be suppressed, and the information redundancy and interference caused by the large number of input bands can be avoided;
- The spatial resolution of the DPC imagery is quite low (3.3 km × 3.3 km), hard to distinguish between cloud and the underlying surface, and the texture is not discernible, particularly in areas where cloud and snow coexist. This means that its spatial information is extremely limited. Hence, in this case, we collected rich spectral data as well as polarization data to provide more precise cloud detection.
- Q and U images already contain all the polarization information, and the solar zenith angle information is already included in the reflectance data, so there is no need to use the polarization reflectance data in our dataset.

According to the related research, the nine angles of DPC imagery contain a considerable amount of information, and the information contained within each of the angles complements one another and can help increase the accuracy of cloud detection [41–43]. Thus, all nine angles of DPC imagery are utilized to compensate for this lack of spatial information, thereby improving cloud detection accuracy. In order to more effectively extract the features of each angle, we consider angle as the third dimension in the image and then use 3D convolution to extract the spatial-angle features. We set spectral to the channel dimension. In the proposed MFCD-Net, the input data is composed of three modalities (including reflectance image, polarization image Q, and polarization image U), and the output is binary label image (including two classes, cloud and non-cloud).

### 3.3. Data Processing

According to the requirements of the experiments, we split 12 images into a training dataset and the other 2 images into a validation dataset. However, due to the DPC's imaging mode's characteristics, each image's effective zones are limited (shown in Figure 2). As a result, before using data, we must delete invalid filling zones and edge zones lacking angle information (as shown in Figure 3) to reduce data contamination. Coupled with the hardware device performance constraints, it is necessary to reduce the size of the input images. We selected the effective zones from the images in our research by going through a series of screening steps and obtained image blocks that are 32 × 32 pixels in size. Table 2 provides the summarized information of the dataset. Due to the limited amount of experimental data, our experiments use the validation set when testing experimental precision and do not create additional test datasets. Meanwhile, for the qualitative analysis of our experiments, we cropped five representative examples of size 160 × 160 in the two large images of the validation dataset as test images.

**Table 2.** Detailed information of dataset.

| Dataset | Image Type | Block Size | Number of Blocks |
|---|---|---|---|
| Training dataset | Reflectance image | $32 \times 32 \times 9 \times 8$ | 21,164 |
| | Polarization image Q | $32 \times 32 \times 9 \times 3$ | |
| | Polarization image U | $32 \times 32 \times 9 \times 3$ | |
| | Label image | $32 \times 32$ | |
| Validation dataset | Reflectance image | $32 \times 32 \times 9 \times 8$ | 2849 |
| | Polarization image Q | $32 \times 32 \times 9 \times 3$ | |
| | Polarization image U | $32 \times 32 \times 9 \times 3$ | |
| | Label image | $32 \times 32$ | |

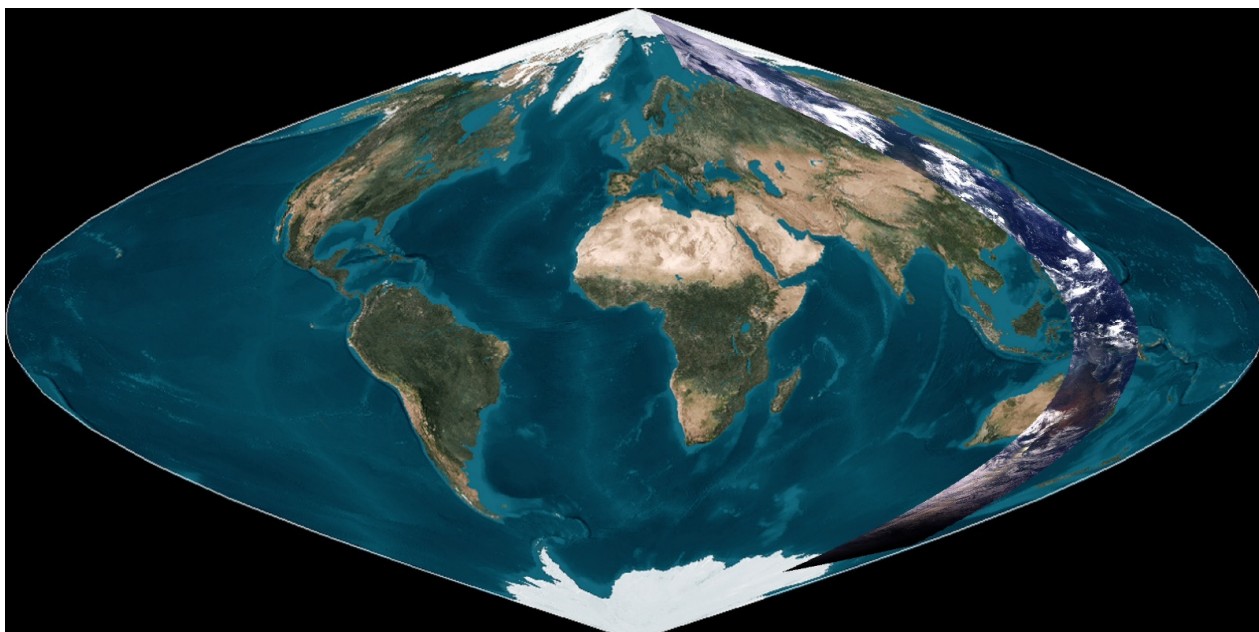

**Figure 2.** DPC overview imagery on 21 March 2017. The size of the whole image is 12,168 × 6484. The striped zone on the right is the valid observation area, and the rest is the invalid filled zone. This striped image is obtained by pseudo-color synthesis of the reflectance image of the third angle of 670 nm band, 565 nm band, and 490 nm band.

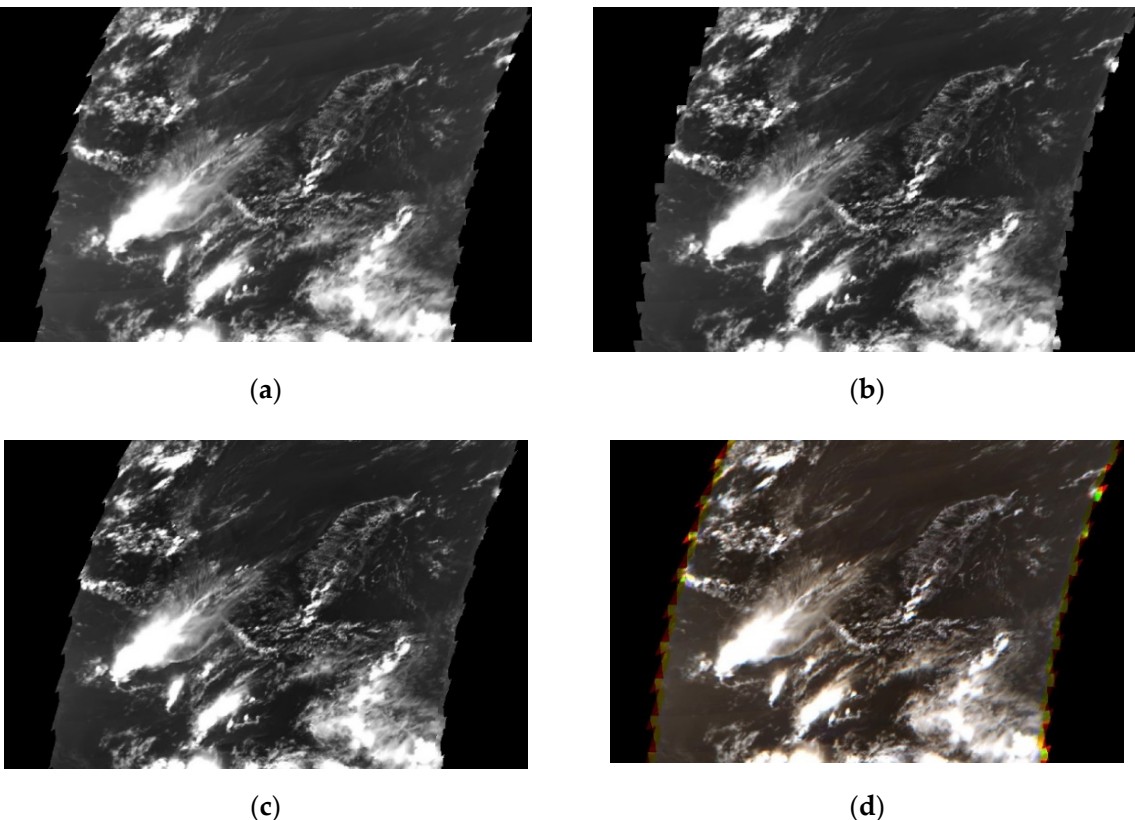

(**a**)　　　　　　　　　　　　　　　　　　　　　　　(**b**)

(**c**)　　　　　　　　　　　　　　　　　　　　　　　(**d**)

**Figure 3.** Examples of DPC imagery from different angles. (**a**–**c**) DPC imagery of $1^h$-angle, $2^h$-angle and $3^{th}$-angle, (**d**) pseudo-color composite image of $1^h$-angle, $2^h$-angle and $3^{th}$-angle. The pseudo-color image shows that the edge zone of the images from different angles does not overlap. These areas lacking certain angle information are invalid edge areas.

## 4. Proposed Method

The purpose of this research is to use deep learning technology for pixel-level DPC imagery cloud detection. To accomplish this task, we propose a 3D multimodal fusion network based on cross attention (MFCD-Net). In this part, we first introduce the overall overview and specific framework of MFCD-Net and then focus on the cross-attention (CA) fusion module.

### 4.1. Model Overview

As shown in Figure 4, our network framework is an end-to-end network with three inputs and one output (cloud mask label). Specifically, our input data consists of three modalities: reflectance image R ($32 \times 32 \times 9 \times 8$), polarization image Q ($32 \times 32 \times 9 \times 3$), and polarization image U ($32 \times 32 \times 9 \times 3$). The output of the network is a cloud mask label image. In order to extract the feature of variation between angles, we use angle as the third dimension of 3D convolution. To highlight the spectral information of important bands, we set the spectral band as the channel dimension, thus using Senet to attach different weights to different bands. The key point of this model is to use the CA module to synergistically fuse the spectral-spatial-angle features of reflectance images and the polarization features of polarization images.

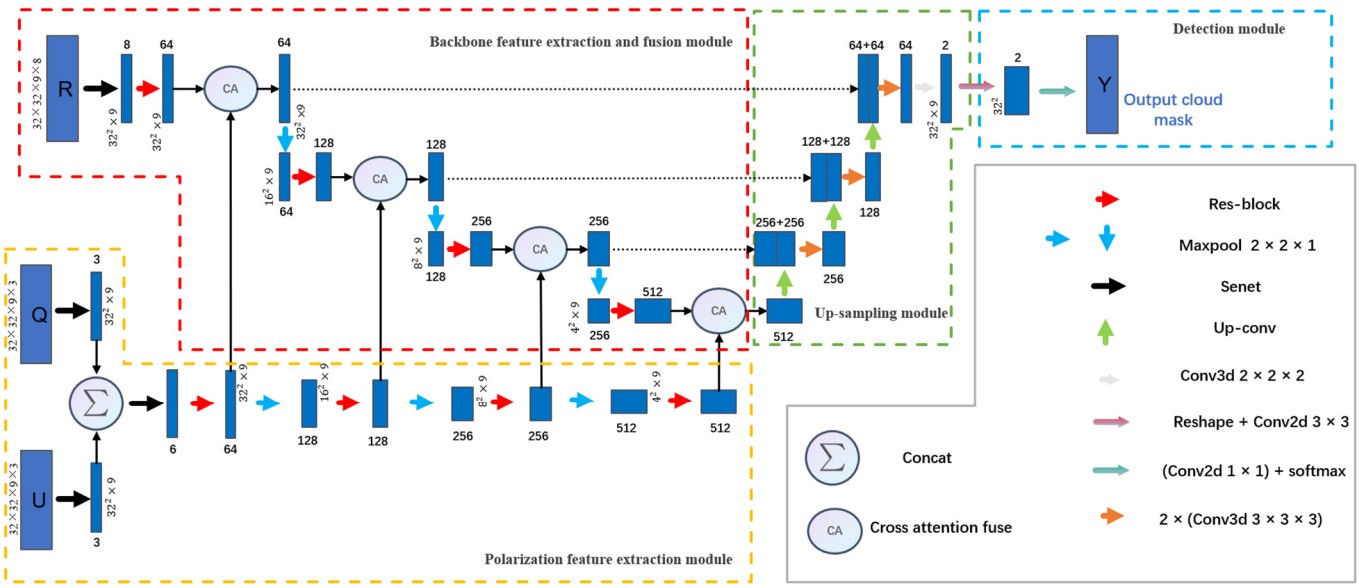

**Figure 4.** Schematic of MFCD-Net.

### 4.2. Network Architecture

The accuracy of feature extraction directly affects the accuracy of the final classification. For very deep networks, networks with a residual structure are easier to train and optimize and achieve impressive performance in a variety of visual tasks [44]. According to the structure of 3D-ResNet-18 [45], we design a Res-block (shown in Figure 5) with two skip connections. The Res-block replaces the ordinary convolutional in the feature extraction process of our model and enhances the feature extraction performance.

As shown in Figure 4, the MFCD-Net architecture is an encoder–decoder structure similar to 3D-UNet. It is divided into four parts: the polarization feature extraction module (PFE), the backbone feature extraction and fusion module (BFEF), the up-sampling module, and the detection module. Each feature map in the model except the detection module is a 4D array with the size of height $\times$ width $\times$ depth $\times$ channel, where height and width are spatial dimensions, depth is angle dimension, and the channel is spectral dimension.

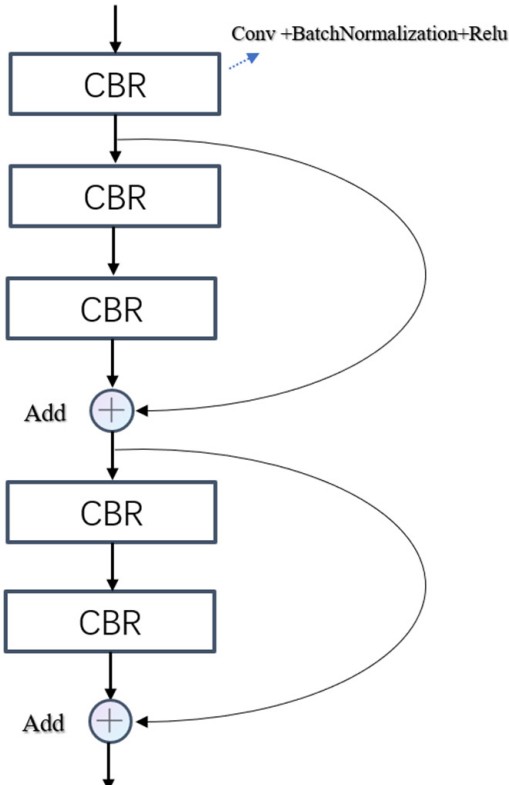

**Figure 5.** The structure of Res-Block. CBR represents convolution + batchnomlization + relu.

In the overall MFCD-Net, the PFE module is used to extract the polarization features of polarization images Q and U at different levels. In the BFEF module, we fuse these polarization features using the CA module at different scales with the spatial-angle-spectral features extracted from the reflectance images. Next, we decode these fused high-level features in the up-sampling module. Finally, the fused features are output as cloud masks through the detection module. These modules are discussed as follows.

- **Polarization feature extraction module.** In the PFE module, we first utilize the concatenation operation to fuse the features of polarization components Q and U. Meanwhile, we use theSenet (shown in Figure 6) before and after the concatenation operation to highlight useful bands and suppress inefficient bands. Then, to extract different levels of polarization features for the fused polarization features, we employ Res-block and the max-pooling operation to extract different levels of polarization features in the extraction structure to adequately fuse polarization features with backbone features (spatial-angle-spectral features). The number of channels in different stages is set to 64, 128, 256, and 512, respectively. Max-pooling operation parameter set to (2,2,1). The size of the feature map at different stages is $32 \times 32 \times 9$, $16 \times 16 \times 9$, $8 \times 8 \times 9$, and $4 \times 4 \times 9$. The convolution kernel is set to size $3 \times 3 \times 3$, and the stride is 1.
- **Backbone feature extraction and fusion module.** The structure of the BFEF module is similar to that of the PFE module, which is based on Res-block and max-pooling. In addition, Senet is also utilized to highlight the information of important bands for the input original feature. Furthermore, the BFEF module additionally contains a step for feature fusion. We fuse the polarization features with the backbone features on the equal stage to reduce information loss as much as possible. The number of channels in each stage, the max-pooling parameter, convolution parameter, and the feature map size are all the same as those in the PFE model.

- **Up-sampling module.** The decoder part of the entire network is the up-sampling module. The up-sampling module's purpose is to restore the size of the high-level fused features output by the BFEF module to that of the original input features. As shown in Figure 4, after each up-sampling, a convolution is applied to adjust the number of channels, and we represent this operation as up-conv. The size of up-sampling and the convolution kernel is set to (2,2,1), (2,2,2). After each up-conv, the feature maps of each stage in the BFEF are merged with the feature maps in the up-sampling module via a skip connection to enhance the fusion of features.
- **Detection module.** The detection module's purpose is to convert the fused feature maps into cloud masks. Since the feature maps input to the detection module are 3D, and the cloud mask is 2D, we first need to reshape them to 2D. Specifically, in order not to change the spatial representation of the feature maps, the reshaping operation aggregates the third dimension and the channel dimension of the 3D feature maps into the channel dimension and does not change the content of the spatial dimension. That is followed by a convolution layer with a $1 \times 1$ kernel, two filters (indicating two classes, cloud and non-cloud), and a softmax activation function. In the end, the classification result of cloud and non-cloud at the pixel level will be obtained.

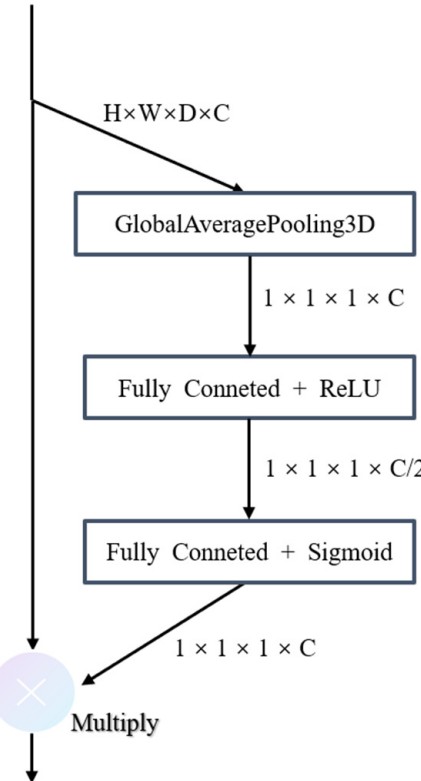

**Figure 6.** The structure of Senet. Unlike the regular 2D Senet, 3D average pooling is used to aggregate feature maps because the feature maps and input data in the network are three-dimensional.

### 4.3. Cross Attention Fusion

In multimodal learning, feature extraction for different modalities is often performed separately rather than jointly using features from both modalities. This leads to the omission of some important shared high-level features in both modalities. Moreover, a simple concatenation fusion would make the features of different modalities significantly unbalanced. In general, attention-based learning focuses on a single modality; thus, only features that are similar are highlighted. The goal of the CA module is to take the polarization modality's attention map and utilize it to improve the representation of the reflectance modality. Simultaneously, the reflectance modality's attention map is multiplied with reflectance features to obtain the self-attentional feature map. It can reduce the information

loss of reflectance modality in the fusion process. In order to enhance the fusion effect, the whole process was carried out successively in the channel dimension and the spatial-angle dimension, and the channel attention and spatial-angle attention were used to generate the attention maps, respectively.

As shown in Figure 7, given a reflectance modality feature map $R \in \mathbb{R}^{H \times W \times D \times C}$ and polarization modality feature map $P \in \mathbb{R}^{H \times W \times D \times C}$ as inputs, the CA module output the fused feature map $F'' \in \mathbb{R}^{H \times W \times D \times C}$. The overall process can be summarized as follows:

$$
\begin{aligned}
F' &= f^{3 \times 3 \times 3}([R \otimes M_C(R) \, ; \, R \otimes M_C(P)]) \\
&= f^{3 \times 3 \times 3}\left(\left[F'_{self} \, ; \, F'_{cross}\right]\right)
\end{aligned}
\tag{3}
$$

$$
\begin{aligned}
F'' &= f^{3 \times 3 \times 3}([F' \otimes M_{SA}(R) \, ; \, F' \otimes M_{SA}(P)]) \\
&= f^{3 \times 3 \times 3}\left(\left[F''_{self} \, ; \, F''_{cross}\right]\right)
\end{aligned}
\tag{4}
$$

where $F'_{self} \in \mathbb{R}^{H \times W \times D \times C}$ and $F'_{cross} \in \mathbb{R}^{H \times W \times D \times C}$ represent the self-attention and cross-attention feature map in channel dimension, respectively, $F' \in \mathbb{R}^{H \times W \times D \times C}$ is the fused feature map in channel dimension, $f^{3 \times 3 \times 3}$ means a 3D convolution operation with the filter size of $3 \times 3 \times 3$, $\otimes$ means the element-wise multiplication, $M_C$ means the channel attention module, $M_{SA}$ means the spatial-angle attention module, and $F''_{self} \in \mathbb{R}^{H \times W \times D \times C}$ and $F''_{cross} \in \mathbb{R}^{H \times W \times D \times C}$ represent the self-attention and cross-attention feature map in spatial-angle dimension, respectively.

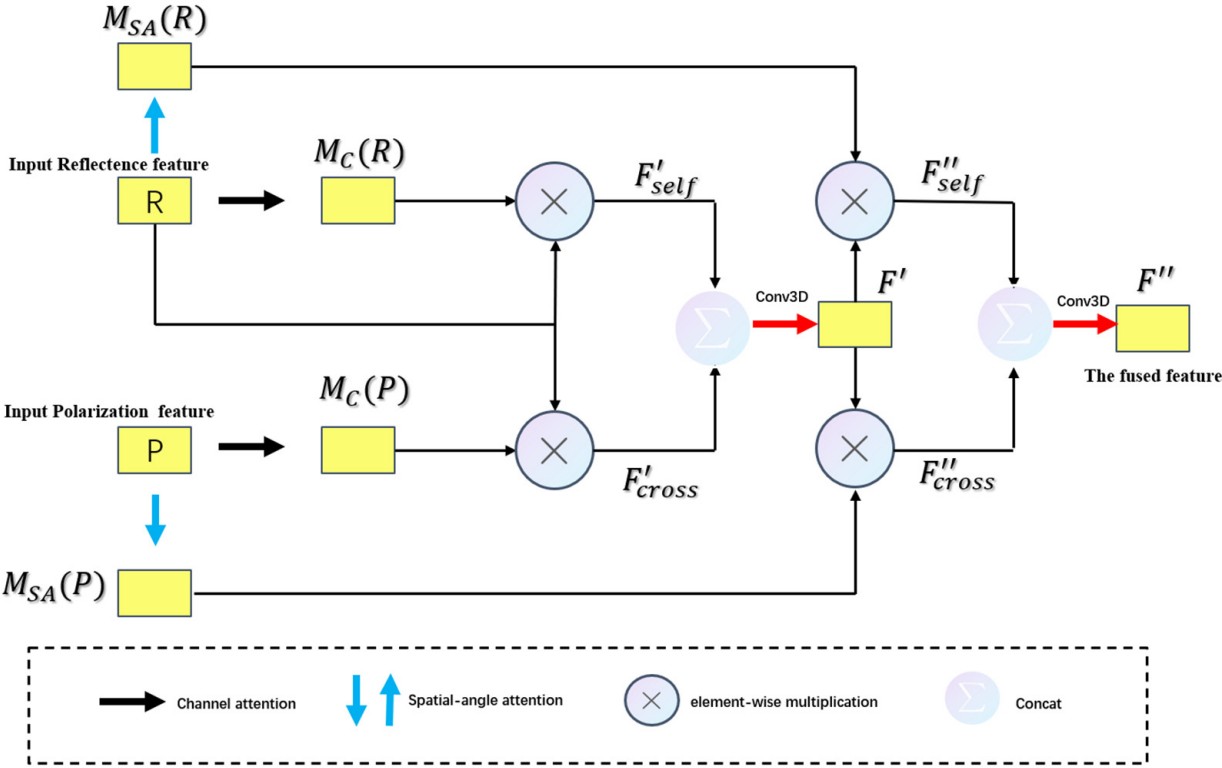

**Figure 7.** Detailed structure of CA module.

Channel attention module $M_C$ and spatial-angle attention module $M_{SA}$ are two important parts of the CA module, which generate attention weight in channel dimension and spatial-angle dimension, respectively. Figures 8 and 9 depict the computation process of each attention map. The following describes the details of each attention module.

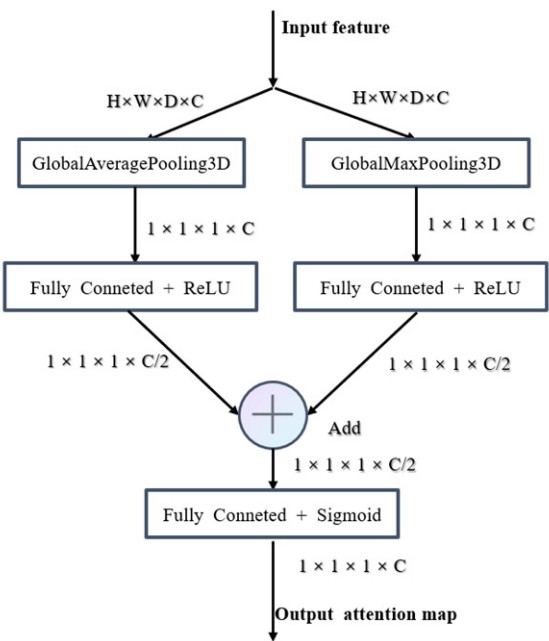

**Figure 8.** Detailed structure of channel attention module.

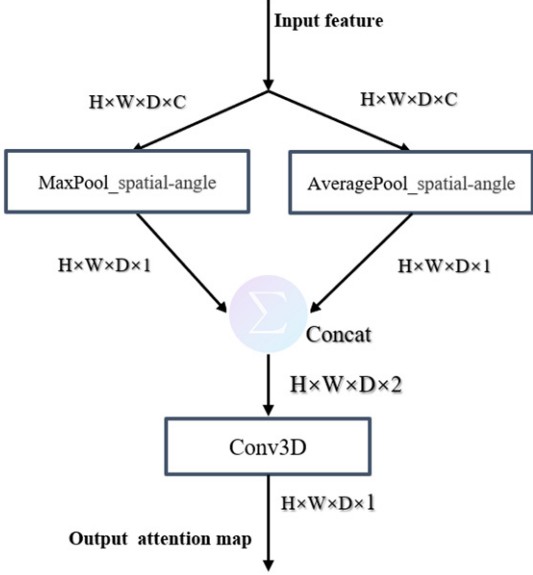

**Figure 9.** Detailed structure of spatial-angle attention module.

For aggregating spatial information, average-polling has been proven to be effective [46]. However, some studies have demonstrated that max-pooling can obtain another important feature, and using both average-polling and max-pooling features simultaneously can obtain better performance [47]. In contrast, what we need in the process of obtaining channel weights is the aggregation of spatial-angle three-dimensional information. Here we utilize average-pooling and max-pooling simultaneously to generate two different feature descriptors: $F_{avg}^C \in \mathbb{R}^{1\times1\times1\times C}$ and $F_{max}^C \in \mathbb{R}^{1\times1\times1\times C}$. The detailed operation is computed as:

$$
\begin{aligned}
M_c(F) &= W_2(W_1(GAP(F)) + W_1(GMP(F))) \\
&= W_2\left(W_1\left(F_{avg}^C\right) + W_1\left(F_{max}^C\right)\right)
\end{aligned}
\tag{5}
$$

where $F \in \mathbb{R}^{H \times W \times D \times C}$ is the input feature map, $GAP$ and $GMP$ denote the GlobleAverage-Pooling3D and GlobleMaxPooling3D operation, respectively, $W_1 \in \mathbb{R}^{1 \times 1 \times 1 \times C/2}$ denotes the full connection layer with the ReLU activation function, $W_2 \in \mathbb{R}^{1 \times 1 \times 1 \times C}$ denotes the full connection layer with the sigmoid function, and $M_c(F) \in \mathbb{R}^{1 \times 1 \times 1 \times C}$ is the output weight feature vector.

For 2D feature maps, a spatial attention map is generally used to express the importance of each spatial pixel. In this paper, we generate the spatial-angle attention map to represent the degree of importance of each point in a 3D feature map. We compute the spatial-angle attention by applying average-pooling and max-pooling operations along the channel axis and concatenating them to generate an efficient feature descriptor. Convolution is applied to the concatenated feature descriptors in order to generate the spatial-angle attention map, which indicates where to emphasize or suppress information. The detailed operation is computed as:

$$
\begin{aligned}
M_{SA}(F) &= f^{3 \times 3 \times 3}([AvgPool(F); MaxPool(F)]) \\
&= f^{3 \times 3 \times 3}\left(\left[F_{avg}^{SA}; F_{max}^{SA}\right]\right)
\end{aligned}
\tag{6}
$$

where $F \in \mathbb{R}^{H \times W \times D \times C}$ is the input feature map, $f^{3 \times 3 \times 3}$ represents a convolution layer with the filter size of $3 \times 3 \times 3$ and the sigmoid function, $F_{avg}^{SA} \in \mathbb{R}^{H \times W \times D \times 1}$ and $F_{max}^{SA} \in \mathbb{R}^{H \times W \times D \times 1}$ are two 3D maps after aggregating channel information of a feature map by using two pooling operations $AvgPool$ and $MaxPool$, and $M_{SA}(F) \in \mathbb{R}^{H \times W \times D \times 1}$ is the output weight feature map.

## 5. Experiments

In this section, we comprehensively evaluate the proposed MFCD-Net on DPC imagery. Specifically, we first describe the experimental setup and the evaluation metrics. Then, we evaluate the performance of our proposed MFCD-Net qualitatively and quantitatively. Third, we further investigate the performance of the CA module, Senet, and Res-block. Finally, we demonstrate the effectiveness of our method of DPC imagery selection and processing.

### 5.1. Experimental Settings
#### 5.1.1. Training Details

All networks were trained under the Keras framework and optimized by the Adam algorithm [48]. The proposed MFCD-Net is trained in an end-to-end manner. The learning rate starts from $10^{-5}$, and is then dynamically changed by the 'Reduce LR On Plateau' function. Specifically, when the validation-loss does not decrease for three epochs, the learning rate will drop to the original 0.8. The whole training process has a total of 100 epochs, and the training will end when the validation-loss does not decrease for 30 epochs. The loss function used in the experiment is the cross-entropy loss function. Comparison methods are trained with the same settings as the MFCD-Net.

#### 5.1.2. Evaluation Metrics

Such commonly used semantic segmentation metrics as overall accuracy ($OA$), producer accuracy ($PA$), user accuracy ($UA$), and $MIoU$ (Mean Intersection over Union) have been employed as evaluation metrics to examine the performance of the cloud detection methods. The formulas for calculating these evaluation indicators are as follows:

$$
OA = \frac{TP + TN}{TP + TN + FP + FN}
\tag{7}
$$

$$
PA = \frac{TP}{TP + FN}
\tag{8}
$$

$$
UA = \frac{TP}{TP + FP}
\tag{9}
$$

$$MIoU = \frac{(IoU_P + IoU_N)}{2} = \frac{\frac{TP}{TP+FP+FN} + \frac{TN}{TN+FN+FP}}{2} \qquad (10)$$

where TP, TN, FP, and FN denote the number of correctly identified cloud pixels, the number of correctly identified non-cloud pixels, the number of incorrectly identified cloud pixels, and the number of incorrectly identified non-cloud pixels, respectively, $IoU_P$ represents the $IoU$ (Intersection over Union) of cloud pixel, and $IoU_N$ represents the $IoU$ of non-cloud pixels.

### 5.1.3. Data Augmentation

The use of data augmentation techniques can, to a certain extent, avoid the overfitting problem and improve the generalization ability of the model [49]. The MFCD-Net is based on three-dimensional convolution, with a considerable number of parameters and a large demand for training data. Because of the poor resolution of DPC imagery, obtaining a large number of label images is challenging; hence the dataset we use is limited and cannot match the network model's requirements. We have performed data enhancement operations on the training data and label, such as vertical flipping, horizontal flipping, and diagonal mirror flipping, to improve the robustness of the network model.

### 5.2. Comparative Experiment of Different Methods

The compared methods include the classical semantic segmentation method: FCN, U-Net, Seg-Net, PSP-Net, and DeepLab-V3. In addition, a comparison is made with some advanced cloud detection methods: deformable contextual and boundary-weighted network (DABNet) [37], lightweight cloud detection network (LWCDnet) [36], and cloud detection UNet (CDUNet) [35]. In order to compare objectively and effectively, the parameters of the experiment are kept consistent. Since the above comparison methods are based on 2D-CNN, we modified the reflectance image and polarization images into 2D form to form multi-channel data. The block size of the input image in 2D-CNN-based networks is $32 \times 32 \times 126$.

Figure 10 shows the experiment's qualitative comparison results on five examples from our dataset. These examples include different backgrounds, such as sea and land regions and special ocean flare regions. In addition, thick and thin cloud scenarios are included. It can be seen that the proposed MFCD-Net has the fewest misclassified pixels in all cases, showing that it has the best capacity to distinguish cloud pixels. It should be noted that our method has significantly fewer misclassified pixels in the edge region than other methods, proving that it can solve the problem of difficult cloud boundary identification well.

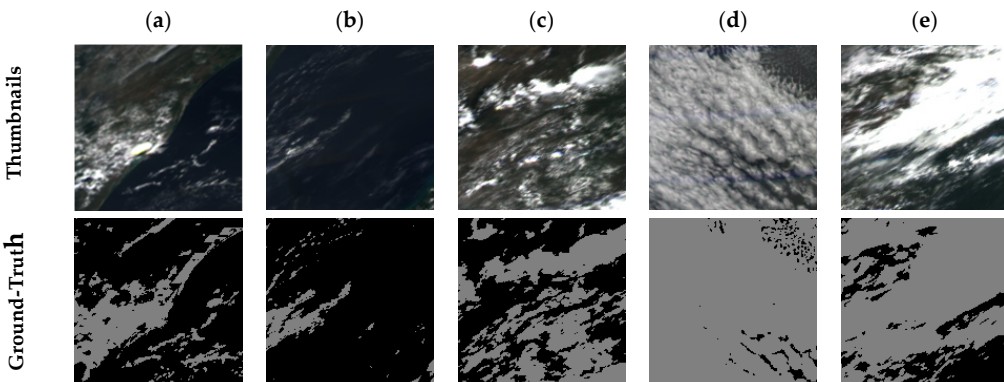

**Figure 10.** *Cont.*

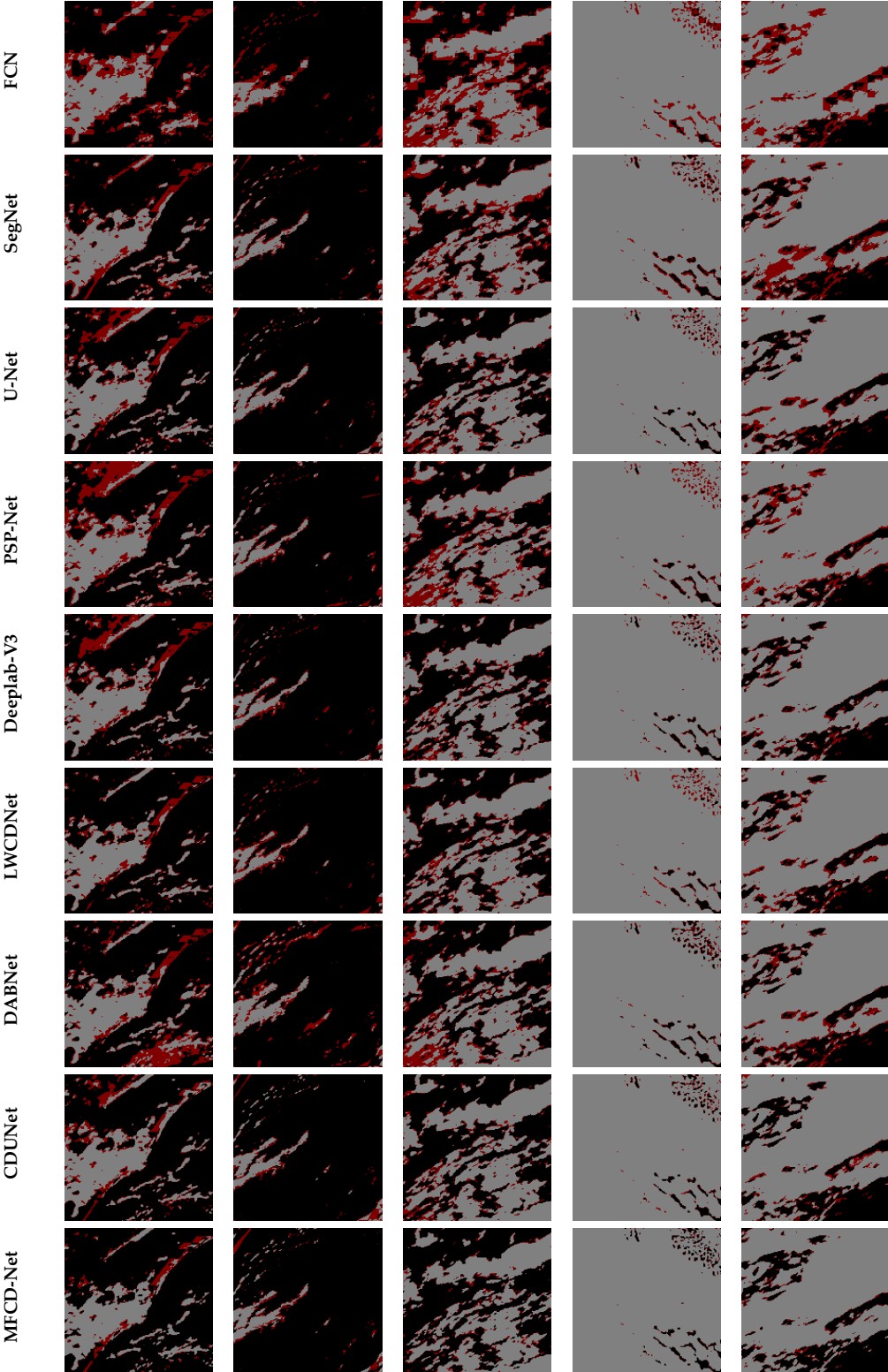

**Figure 10.** Experiment's qualitative comparison results. The size of all images is 160 × 160. The thumbnails are the RGB pseudo-color composition of the third angle reflectance image of 670 nm band, 565 nm band, and 490 nm band. (**a**–**e**) are examples of five different situations; (**a**) is the sea–land mixed case, (**b**) is the thin cloud case in the ocean region, (**c**) is the land region, (**d**) is the thick cloud case, and (**e**) is the flare region. Gray represents cloud, black represents non-cloud, and red represents misclassification result.

Quantitative comparison results are shown in Table 3. In all the evaluation metrics, our method achieves the best performance, especially MIoU, which is improved by at least 3.38% compared with other methods. The results indicate that the proposed MFCD-Net outperforms 2D-CNN-based comparison methods, showing the superiority of 3D convolution in feature extraction of multi-dimensional data. Along with the proposed MFCD-Net, the U-net and CDUNet achieved 92.95% and 93.82% in terms of OA, respectively, demonstrating that the U-shaped encoder–decoder structure has a strong ability to extract the features of the multi-channel data. This is why we choose an encoder–decoder structure similar to U-net in the proposed MFCD-Net.

**Table 3.** Cloud detection accuracy of different methods (%).

| Method | OA | UA | PA | MIoU |
|---|---|---|---|---|
| FCN | 86.45 | 89.24 | 88.84 | 75.06 |
| Seg-net | 89.95 | 90.11 | 92.81 | 80.41 |
| U-net | 92.95 | 93.30 | 95.17 | 86.24 |
| PSP-Net | 91.18 | 94.38 | 91.58 | 82.87 |
| Deeplab-V3 | 92.76 | 93.66 | 94.56 | 85.88 |
| LWCDNet | 92.37 | 91.78 | 95.67 | 85.26 |
| DABNet | 92.36 | 92.46 | 95.02 | 85.20 |
| CDUNet | 93.82 | 94.37 | 95.56 | 87.82 |
| MFCD-Net | **95.74** | **96.38** | **96.96** | **91.20** |

*5.3. Ablation Study*

To demonstrate the advancement and effectiveness of our designed network, we implemented ablation experiments on the DPC dataset for the Res-block, Senet, and CA modules. We list the evaluation performance of the backbone network, as shown in Table 4. The experiments have been carried out on both unimodal data (input reflectance data only, represented as (R) in Table 4) as well as multimodal data (input both reflectance and polarization data, represented as (R+P) in Table 4) to eliminate the effect of data on experiments.

**Table 4.** The cloud detection accuracy (%) for ablation study.

| Method. | OA | UA | PA | MIoU |
|---|---|---|---|---|
| 3D-UNet (R) | 93.69 | 94.66 | 90.93 | 87.95 |
| 3D-UNet (R+P) | 94.56 | 93.76 | 94.02 | 89.57 |
| 3D-UNet + Senet(R) | 94.18 | 95.63 | 95.31 | 88.15 |
| 3D-UNet + Senet(R+P) | 95.03 | 96.60 | 95.64 | 89.73 |
| 3D-UNet + Senet + Res-block (R) | 95.00 | 96.36 | 95.86 | 89.72 |
| 3D-UNet + Senet + Res-block (R+P) | 95.31 | **97.01** | 95.74 | 90.29 |
| 3D-UNet + Senet + Res-block + CA | **95.74** | 96.38 | **96.96** | **91.20** |

Our network structure is improved based on the 3D-UNet. In order to highlight the important band information of the input data, Senet is used to attach different weights to the features of different bands. It can be seen that the Senet improves the performance from 93.69% to 94.18% in the unimodal case and from 94.56% to 95.03% in the multimodal case in terms of OA. Based on the incorporation of Senet, we further introduce Res-block to replace the normal CBR-block (*Conv + BatchNormalization + Relu*) in the feature extraction process in 3D-Unet and thus steadily improve the model by deepening the number of network layers. The experimental results show that the use of Res-blocks has a significant improvement in the performance of the model in all evaluation metrics. The CA module is the core part of MFCD-Net. Different from the commonly used feature concatenation fusion method, it highlights the reflectance feature by an attentional map of the polarization feature, synergistically fusing the multimodal feature. The ablation networks without the CA module use a feature concatenation fusion method in the case of multimodal data. It

can be seen that MFCD-Net (3D-UNet + Senet + Res-block + CA) achieves the highest performance in terms of OA, MIoU, and PA compared with other ablation networks.

### 5.4. Selection of Dataset

Considering DPC imagery's multi-spectral, multi-angle, and multi-polarization properties, the selection of band and angle and the use of polarization images are very important steps. We conducted a series of experiments to illustrate the effectiveness of our data selection strategy of separating all data into three modalities and then performing multimodal fusion.

First, in the case of using multi-angle data, we respectively input the R (reflectance) image, Q (Stokes vector Q) image, and U (Stokes vector U) image of each band into the single modality MFCD-Net network for experiments. Specifically, by altering the input data, we conducted the following experiments to demonstrate the effectiveness of using multi-band combined images: inputting 3-band (including various combinations) R images, 8-band R images, 3-band Q images, and 3-band U images. As shown in Table 5, 8-band R image input outperforms 3-band and single-band R image input. Simultaneously, the performance of 3-band polarization images (Q and U) input is superior to that of single-band polarization image input, demonstrating the effectiveness of using multi-band data input in this research. The detection effect of employing merely R image input, Q image input, or U image input is not as good as the proposed method in this research. This demonstrates that neither the polarization image nor the reflectance image can include all of the necessary information, demonstrating the efficacy of the data use method presented in this study.

**Table 5.** The cloud detection accuracy (%) of different input bands.

| Data | | OA | UA | PA | MIoU |
|---|---|---|---|---|---|
| Single-band | 443 nm | 91.42 | 89.49 | 91.43 | 84.06 |
| | 490 nm | 92.25 | 89.93 | 92.98 | 85.50 |
| | 565 nm | 92.36 | 90.15 | 92.98 | 85.69 |
| | 670 nm | 92.58 | 90.58 | 92.98 | 86.06 |
| | 763 nm | 91.89 | 89.81 | 92.22 | 84.87 |
| | 765 nm | 91.56 | 90.38 | 90.67 | 84.27 |
| | 865 nm | 92.96 | 90.93 | 91.73 | 85.49 |
| | 910 nm | 91.47 | 90.22 | 90.67 | 84.12 |
| | Q490 nm | 78.85 | 75.19 | 78.32 | 64.89 |
| | Q670 nm | 79.86 | 76.52 | 78.32 | 65.99 |
| | Q865 nm | 80.21 | 76.88 | 79.33 | 66.74 |
| | U490 nm | 79.25 | 75.31 | 79.33 | 65.46 |
| | U670 nm | 80.21 | 76.95 | 79.20 | 66.74 |
| | U865 nm | 82.25 | 79.86 | 80.32 | 69.60 |
| 3-band | Q490-Q670-Q865 | 85.46 | 84.34 | 82.63 | 74.33 |
| | U490-U670-U865 | 86.52 | 86.03 | 83.17 | 75.95 |
| | 443-490-865 | 93.56 | 92.38 | 93.20 | 87.78 |
| | 490-670-865 | 94.35 | 94.73 | 92.42 | 89.16 |
| | 443-670-865 | 93.86 | 94.26 | 91.78 | 88.27 |
| | 763-765-910 | 93.04 | 93.12 | 91.06 | 86.82 |
| | 565-765-910 | 93.53 | 92.86 | 92.56 | 87.71 |
| 8-band | | **95.00** | **96.36** | **95.86** | **89.72** |

Next, a few experiments on multi-angle and single-angle cases were carried out under the adjusted single modality input MFCD-Net network to demonstrate the importance of multi-angle data. Because the size of the third dimension (angle dimension) of the feature maps is 1 in the case of single-angle input, we need to alter the size of the convolution kernel (must be larger than the size of the feature map's third dimension). We added two groups of experiments to increase the credibility of the experiment, input 3-angle data and input 6-angle data, to compare with the approach of input 9-angle data in this paper. Table 6

shows that the $1^h \sim 9^{th}$-angle input achieved the best performance, and the experimental effect improved as input angle numbers rose. This highlights the role of each angle and the effectiveness of using multi-angle data.

Finally, we replace the reflectance image input with the radiance image input in this approach to show that the reflectance image has more information than the radiance image discussed in Section 3. The detection performance of the approach utilizing radiance input is not as good as that of the method in this paper using reflectance input, as shown in Table 7. As this experiment illustrates, reflectance data can provide more information than radiant brightness data. As a consequence, in our dataset, we used reflectance data rather than radiance data.

**Table 6.** The cloud detection accuracy (%) of different input angles.

| Data | OA | UA | PA | MIoU |
|---|---|---|---|---|
| $1^h$-angle | 87.53 | 88.24 | 91.99 | 78.11 |
| $2^h$-angle | 87.79 | 82.21 | 92.56 | 78.18 |
| $3^{th}$-angle | 87.63 | 83.66 | 95.60 | 77.28 |
| $4^{th}$-angle | 88.56 | 83.02 | 93.35 | 79.42 |
| $5^{th}$-angle | 88.91 | 89.20 | 93.22 | 78.92 |
| $6^{th}$-angle | 89.06 | 83.86 | 93.35 | 80.21 |
| $7^{th}$-angle | 89.73 | 88.11 | 95.53 | 80.50 |
| $8^{th}$-angle | 88.86 | 83.52 | 93.35 | 79.90 |
| $9^{th}$-angle | 88.54 | 88.49 | 90.83 | 78.36 |
| $1^h \sim 3^{th}$-angle | 92.03 | 93.06 | 93.06 | 85.01 |
| $4^{th} \sim 6^{th}$-angle | 92.85 | 94.90 | 88.68 | 86.42 |
| $7^{th} \sim 9^{th}$-angle | 92.99 | 94.49 | 89.45 | 86.69 |
| $1^h \sim 6^{th}$-angle | 93.89 | 94.88 | 95.56 | 88.55 |
| $3^{th} \sim 9^{th}$-angle | 94.12 | 95.63 | 90.93 | 88.72 |
| $1^h \sim 9^{th}$-angle | **95.00** | **96.36** | **95.86** | **89.72** |

**Table 7.** The cloud detection accuracy (%) for evaluating reflectance image.

| Data | OA | UA | PA | MIoU |
|---|---|---|---|---|
| Radiance image input | 95.01 | 95.87 | 95.52 | 89.56 |
| Reflectance image input | **95.74** | **96.38** | **96.96** | **91.20** |

## 6. Conclusions

DPC on GF-5 satellite is the only on-orbit polarization detector in the world at present, which can provide multi-spectral, multi-angle and multi-polarization observations. Cloud detection is the cornerstone of DPC imagery application. In this paper, we propose a neural network (MFCD-Net) to extract cloud masks from DPC imagery. MFCD-Net has three key innovations: (1) It extracts spatial-angular-spectral information by 3D convolution; (2) it uses the cross-attention fusion mechanism to fuse polarization information; (3) its overall structure is similar to 3D-Unet, and at all feature levels, the features of the same level are fully extracted and fused. The effectiveness of the proposed MFCD-Net is verified with the experimental results compared with various methods.

**Author Contributions:** K.G. designed and completed the experiments and drafted the manuscript. J.Z. (Jingjing Zhang) and T.L. provided the research ideas and modified the manuscript, L.X., M.Z. and J.Z. (Jinqin Zhong) put forward the improvement suggestions for the experiment and the manuscript. W.X. and X.S. provided the source data and data processing principles. All authors assisted in writing and improving the paper. All authors have read and agreed to the published version of the manuscript.

**Funding:** This research was funded by the Key Laboratory of Optical Calibration and Characterization, Chinese Academy of Sciences Open Research Foundation, and the funder is Jingjing Zhang; Anhui Provincial Natural Science Foundation of China (No. 1908085J25), and this funder is Teng Li.

**Data Availability Statement:** Not applicable.

**Acknowledgments:** The authors want to thank the Key Laboratory of Optical Calibration and Characterization, Chinese Academy of Sciences for supporting the experimental data. In addition, we are grateful to the Key Laboratory of Intelligent Computing and Signal Processing of Ministry of Education, School of Electrical Engineering and Automation, Anhui University, who supported hardware devices for this research.

**Conflicts of Interest:** The authors declare no conflict of interest.

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
