# Peer review of "MFCD-Net: Cross Attention Based Multimodal Fusion Network for DPC Imagery Cloud Detection"

_remotesensing, doi:10.3390/rs14163905_

Round 1
Reviewer 1 Report
The manuscript MFCD-Net:Cross Attention based Multimodal Fusion Net-2 work for DPC imagery Cloud Detection introduces a new method for cloud detection. Its novelty lies both in the concrete application of artificial intelligence methods and in the use of new types of data. The paper is suitable for publication after some modification. In general, I have the following comments:
1. The article uses a number of AI terms that are not familiar to the general reader and are not explained in the article, at least in general terms, and the whole description of the methods is difficult to understand. I suggest explaining the basic terms briefly and simplifying the whole description, including the use of shorter sentences. A detailed description, if needed, would be placed in an appendix or in the accompanying files.
2. The basic explanation of reflectivity, polarization reflectivity, reflectance, polarization components Q, U, channel attention module and other terms also deserve a more detailed explanation/description.
3. The presentation of the results by selected methods needs a justification (statistical tests?) that the differences between the values obtained are significant. The tables often show differences in tenths of a percent and conclude that one method/data is better than the other.
4. May be I am wrong but I did not noticed how training and verification data are prepared/divided. Clear descriptions and number of data should be in the manuscript.
5. The text needs language editing.
Specific comments:
L41 – “the top of the atmosphere” is at about 100 km. What do you mean by your statement?
L43 – I do not understand this part of the sentence.
L172 – Remove dot, please.
L180 – The sentence should be reformulated
L299 – What is FEF?
L414 – Second TN should be FN.
L421 – I do not understand “will be easier”.
L420-426 – In my opinion this paragraph should be rewritten. The explanations are not clearly written or wrong.
Author Response
Dear reviewer:
Thank you for your decision and constructive comments on my manuscript. We have carefully considered the suggestion of Reviewer and make changes. We have tried our best to improve and made some changes in the manuscript.

Reviewer 2 Report
The authors proposed a novel dataset for could detection, MFCD-Net is also proposed for cloud detection. However, many issues exist:
1) The authors claimed to build DPC images dataset. However, no download link is given for it.
2) The contribution of the paper is not illustrated clearly in the introduction part.
3) More recent cloud detection method should be reviewed in the Related work part.
4) In Figure3, it is not formable to use such obscure figures instead of tables.
5) In Section 5.2, the compared methods are too old, such as FCN, Seg-Net, UNet, PSP-Net, DeepLab-V3. It is not fair to compare MFCD-Net with these methods, as they are not designed specially for could detection. More could detection methods in the literature should be compared.
6) Ablation study is missed for your MFCD-Net, such as CA module, SENet module, spatial-angle attention module.
Author Response

(The authors gave the same response as above.)

Reviewer 3 Report
This study proposes a cloud detection algorithm applying the 3D-convolutional neural network(CNN) for a sensor, Directional Polarimetric Camera (DPC), which detects polarization of radiance at several solar wavelengths from several angles. The authors aim to make it to be suitable for imagers with low spatial resolution by compensating for the lack of spatial information. This algorithm consists of some network modules for extracting spatial-angle-spectral information, fusing the polarization features, and decoding the feature maps. Experiments of cloud detection and comparisons to other methods reveal that the proposed algorithm is more capable than those based on 2D-CNN, and suggest that the cross-attention fusion and the Residual Network are effective for better cloud detection performance.
Advanced sensors often need sophisticated cloud detection algorithms for meaningful satellite remote sensing observation. Machine learning (or AI) seems to be applicable to satellite image analysis such as cloud detection, but it is still necessary to research how to incorporate the Machine learning techniques into cloud detection methods according to purposes of observation, used sensors, and targets. This study will contribute to improving application of AI to remote sensing and therefore suitable for the journal. I think that this manuscript will be sufficient for publication after some minor corrections.
1. Figure 3, It will be better to illustrate as a satellite image with indicating invalid edge area.
2. P9 L297, "The convolution kernel is set to size 3×3×3", Is stride 1?
3. P9 L318, "Detection module", I feel that more explanation of Detection module is needed. How to reshape the 3D feature map to 2D? What are "two filters"?
4. P12 L364, "Sanghyun Woo et al. suggest that ~", cite the article as reference (or personal communication?)
5. P14 L447, describe the procedure of "visual inspection". How many persons carry out the inspection for one image? What images are used (true color RGB only or adding some pseudo RGB images)?
Author Response
Dear reviewer: Thank you for your decision and constructive comments on my manuscript. We have carefully considered the suggestion of Reviewer and make some changes. We have tried our best to improve and made some changes in the manuscript.
Round 2
Reviewer 2 Report
The authors have basically addressed my concerns. Language and expression are suggested to be further polished before final publication.